# Design of a High Voltage Pulse Generator with Large Width Adjusting Range for Tumor Treatment

**Xin Rao [1], Xiaodong Chen [1,2], Jun Zhou [1,\*], Bo Zhang [1] and Yasir Alfadhl [2]**

[1] School of Electronic Science and Engineering, University of Electronic Science and Technology of China, Chengdu 610054, Sichuan, China; raoxinuestc@gmail.com (X.R.); xiaodong.chen@qmul.ac.uk (X.C.); bozhang@uestc.edu.cn (B.Z.)

[2] School of Electronic Engineering and Computer Science, Queen Mary University of London, London E1 4NS, UK; yasir.alfadhl@qmul.ac.uk

[\*] Correspondence: zhoujun123@uestc.edu.cn

**Abstract:** The unique biological effects stimulated by short pulsed electric field have many applications in tumor treatment, such as irreversible electroporation, electrochemotherapy, gene transfection and immune therapy. These biological effects require high voltage pulses with different pulse width in the range from nanoseconds to hundreds of microseconds. To fulfill this requirement, a compact high voltage pulse generator has been designed based on a switchable capacitor array and a SiC MOSFET switching array. The proposed pulse generator has one output channel with an adjustable pulse width from 100 ns to 100 µs, an amplitude range from 0 kV to 2 kV, a repetition rate less than 1.2 kHz and a voltage drop less than 5%. The mechanism of the stacked switches circuit was investigated, in connection with a switchable capacitor array. The introduction of a switchable capacitor array extends the pulse width from nanosecond scale and microsecond scale compared with other similar design methods. The pulse generator has been designed in simulation and implemented in experiment. The developed pulse generator provides a convenient and economical tool for the further studies of the unique biological effects stimulated by different pulsed electric fields for tumor treatment.

**Keywords:** short pulsed electric field; biological application; switchable capacitor array; SiC MOSFET array

## 1. Introduction

The different biological effects stimulated by short pulsed electric field mainly include reversible electroporation (RE), irreversible electroporation (IRE), cell apoptosis, and so on. RE exploits the pulse with the electric field intensity of 10 kV/m to 1 kV/cm and the width of microseconds [1]. By enhancing the electric field intensity of the microsecond pulse over the irreversible electroporation (IRE) breakdown threshold of 1 kV/cm, IRE can kill cells directly through the necrosis [2–4]. Cell apoptosis can be stimulated by the nanosecond pulsed electric field with the electric field intensity of 10 kV/cm to 300 kV/cm and the width of a few to hundreds of nanoseconds [5]. In the applications of tumor treatment, they all possess unique advantages and limitations. RE, being applied in electrochemotherapy, introduces toxic drugs to the cancer cells through enhancing the cell permeabilization with microsecond pulsed electric fields. Though it is a very effective tumor therapy, it still suffers from the usual side effects of chemotherapy [6–8]. In 1991, the first electrchemotherapy trial of squamous cell carcinoma was conducted, in which eight subjects were treated by 4 or 8 pulses with the pulse width of 100 µs, and the clinical complete cure rate was 57% [9]. A typical commercial generator for RE nowadays is the AgilePulse[TM] System of BTX Co., Ltd., Holliston, MA, USA, which provides a pulse width adjusting range of 50 µs–100 ms [10]. IRE, also called Nano Knife, is a novel tumor ablation technique with

the advantages of drug-free and non-thermal treatment, but its treating area is limited (2–3 cm) [2–4]. In 2010, a solid tumor was ablated by 90 pulses with an ultra-short pulse length of 100 microseconds under the real-time monitoring of computed tomography (CT) [11]. The most widely used clinical device for IRE is the Nano Knife System of Angio-Dynamics Co., Ltd, which provides a pulse width adjusting range of 70–100 μs [12]. Cell apoptosis stimulated by the nanosecond pulsed electric field can elicit antitumor immunity and inhibit tumor migration and recurrence, but its local tumor eradication may not be complete [13–15]. In 2013, the first nanosecond pulse stimulation trial of human basal cell carcinoma (BCC) was reported, and 10 BCCs on three subjects were treated by 100–1000 pulses with the pulse width of 100 ns. Seven of the 10 treated lesions were completely free of basaloid cells when biopsied and two partially regressed [16]. The current commercial generator for nanosecond pulse stimulation is the CellFX™ System of Pulse Biosciences Co., Ltd., Hayward, CA, USA, which provides a pulse width adjusting range of 100–300 ns [17]. These tumor therapies being triggered by the pulsed electric field with different pulse width may complement each other and may be combined for an effective tumor treatment. Therefore, in order to realize these studies conveniently and economically, we have proposed to develop a compact high voltage pulse generator with a wide pulse width adjusting range (100 ns to 100 μs).

There are two popular configurations to generate intense short pulsed electric field: capacitor-based discharging configuration and transmission line configuration [18–33]. The mechanism of transmission line configuration is to employ a transmission line as an energy storage unit which is controlled by a fast switch. This leads to the advantages of fast response and the nice rectangularity of pulses, and the disadvantages of the large size and difficulty in width adjusting [18–26]. A typical design in this configuration with narrow width tuning has been achieved by varying the length of the transmission line, with the output pulse width of 10, 40, 60, 92 ns and a peak voltage of 12.5 kV for studying the cell apoptosis [19]. Its improved version, Blumlien transmission line, makes this system more compact and reduces loading requirements. Based on Blumlien transmission line, many pulse generators were developed by some institutes, while the problem of the narrow width tuning still remain to be solved. The group in Old Dominion University developed many nanosecond pulse generators, and one of them is capable of generating the pluses with the peak voltage of 1 kV and a pulse width of 8–60 ns for study the cell apoptosis [21–23]. The group in Chongqing University also developed a pulse generator with the pulse width of 50–100 ns and a maximum amplitude of 2 kV for study the cell apoptosis [24–26].

The capacitor-based discharging system is using a capacitor or a capacitor array as a transmission line as an energy storage unit instead of the transmission line, which has two common types of the MarX circuit and the stacked switches circuit. The mechanism of the MarX circuit is to charge all the discharge capacitors in parallel firstly, and then these capacitors are discharged in cascade, which is controlled by many distributed switches. Based on this mechanism, the MarX circuit is capable of generating intense pulsed electric field, but the problem of the narrow width tuning still exist [27]. On the other hand, the stacked switch circuit, with the switch block implemented by either Insulated Gate Bipolar Transistors(IGBTs) or Metal-Oxide-Semiconductor Field-Effect Transistors (MOSFETs), generates the intense pulses with a wider pulse width adjustable range, while the limitation being a low peak voltage [28–32]. For the biological applications, a typical design based on stacked MOSFET array controlled generates pulses with a peak voltage of 3 kV and a width of 1 μs–5 ms for electroporation [29], and another one based on IGBT produces pulses with a peak voltage of 4 kV and a width of 3 μs–10 ms for electroporation [33]. Vilnius Gediminas Technical University also developed a pulse generator with a pulse width of 3 μs–10 ms and a maximum amplitude of 3.5 kV for electroporation [34]. In addition, there are many commercial generators beside the three above mentioned systems, such as AVRH-2-B pulse generator of Avtech Electrosystems Ltd, Beijing, China, which is capable of generate pluses with peak voltage of 2 kV and a pulse width of 200 ns–2 μs. In view of the above studies, there is still no single pulse generator with a pulse width range of 100 ns–100 μs.

In this paper, we have introduced a switchable capacitor array to extend pulse width range in the capacitor-based discharging system. In addition, a 3 × 2 MOSFET switching array is employed as

a high-voltage high-speed switch, to ensure a clean pulse shape and an adaptation of the low load impedance in some special experiments. Furthermore, because of the mismatch of the load caused by the diverse experimental situations, the current overflow and oscillation may destroy the low voltage controller, and are mitigated by introducing an optic-coupler driver block. The proposed pulse generator is capable of producing the pulses with a repetition rate less than 1.2 kHz, a voltage range of 0 kV–2 kV, and a pulse width range of 100 ns–100 µs, when applied to loads with variable impedance in simulation and in experiment.

## 2. Circuit Design and Testing

### 2.1. Analysis of the Capacitor-Based Discharging Circuit

The capacitor-based discharging circuit is basically a RLC series circuit with a fast switch S, as shown Figure 1. The integral equation describing the discharge process when S is turned on is as follows:

$$\frac{1}{C_0}\int i\,dt + L_0\frac{di}{dt} + Ri = V_0 \tag{1}$$

where, $L_0$ is the parasitic inductance, R is the resistance of the load, and $C_0$ is the capacitance of the discharging capacitor. $V_0$ is the initial voltage on the discharging capacitor. The homogeneous differential equation can be deduced from Equation (1) and initial conditions as follow:

$$L_0\frac{d^2i}{dt^2} + R\frac{di}{dt} + \frac{1}{C_0}i = 0 \qquad i|_{t=0} = 0 \qquad L_0\frac{di}{dt}|_{t=0} = -V_0 \tag{2}$$

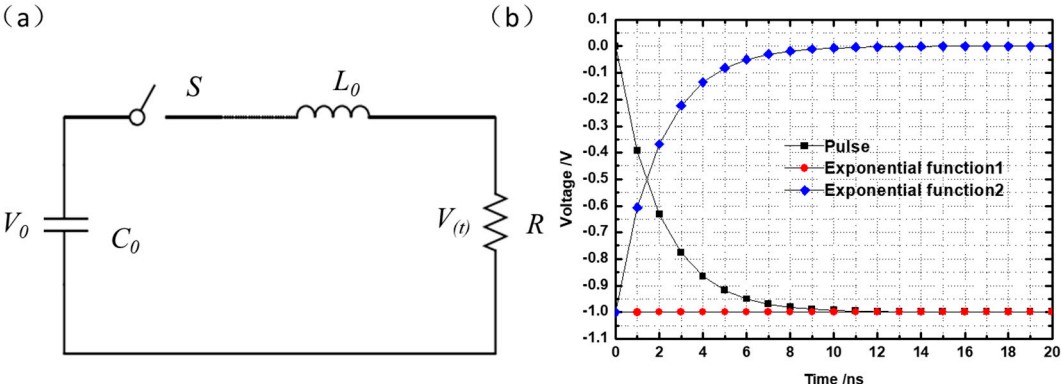

**Figure 1.** (**a**) Schematic diagram of the RLC series circuit simplified from the capacitor-based discharging circuit; (**b**) the voltage on the load and its two decomposed exponential functions: exponential function 1 is $ExpFunc_1$ and exponential function 2 is $ExpFunc_2$ (when $L_0 = 0.1$ µH, $R_0 = 50$ Ω, and $C_0 = 0.5$ µF).

According to the Viete theorem, the two eigenvalues can be easily obtained by solving Equation (2):

$$e_1 = -\frac{R}{2L_0} + \sqrt{\frac{R^2}{4L_0^2} - \frac{1}{L_0C_0}} \qquad e_2 = -\frac{R}{2L_0} - \sqrt{\frac{R^2}{4L_0^2} - \frac{1}{L_0C_0}} \tag{3}$$

With $\alpha$ and $\omega_0$ introduced, the Equation (1) is simplified as follow:

$$e_1 = -\alpha + \sqrt{\alpha^2 - \omega_0^2} \qquad e_2 = -\alpha - \sqrt{\alpha^2 - \omega_0^2} \qquad \alpha = \frac{R}{2L_0} \qquad \omega_0 = \frac{1}{\sqrt{L_0C_0}} \tag{4}$$

According to Equation (4), there are three situations of eigenvalue depended on the positive or negative of $\alpha^2 - \omega_0^2$ When it is negative or zero, this circuit is in either the oscillating state or critical balance, which should be avoided in the design.

Only when $\alpha^2 - \omega_0^2$ is positive, the circuit is suitable for producing stable pulses and the current equation is as follows:

$$i_{(t)} = -\frac{V_0}{(e_1 - e_2)L_0}(e^{e_1 t} - e^{e_2 t}) \tag{5}$$

The voltage $V_{(t)}$ on the load can be easily obtained and divided into two exponential functions, which possess their own unique physical significances, as follows:

$$V_{(t)} = ExpFunc_{1(t)} - ExpFunc_{2(t)} \qquad ExpFunc_{1(t)} = -\frac{V_0 R}{(e_1 - e_2)L_0}e^{e_1 t} \qquad ExpFunc_{2(t)} = -\frac{V_0 R}{(e_1 - e_2)L_0}e^{e_2 t} \tag{6}$$

The time constants (TC) of two exponential functions are the absolute value of the reciprocal value of their eigenvalue as follows:

$$TC_1 = \frac{1}{|e_1|} \qquad TC_2 = \frac{1}{|e_2|} \tag{7}$$

The Figure 1b shows the voltage on the load and its two decomposed exponential components, when the parasitic inductance $L_0$ is 0.1 μH, the load resistance is 50 Ω, and the discharge capacitance $C_0$ is 0.5 μF. The exponential function 1 $ExpFunc_1$ with a huge time constant of 25 μs provides a stable waveform to the pulse up to a few microseconds. The exponential function 2 $ExpFunc_2$ with a time constant of 2 ns determines a rapid rising edge. Together with fast switching on and off, the circuit is capable of producing the high voltage pulses with a tunable pulse width below a few microseconds.

To extend the pulse width adjusting range, an exponential function (1) with a bigger time constant is acquired, which means a large discharge capacitance $C_0$ is required. As shown in Figure 2a, the drop of the voltage on the load decreases with the increase of the discharge capacitor value. And the time constant of exponential function (1) is increased from 25 μs to 1 ms. For generating the pulse with a width of 100 μs with an acceptable voltage drop, one needs a discharge capacitor with capacitance of 20.5 μF at least. In addition, a low resistance of load also leads to a slight voltage drop in exponential function (1) in a similar way, and this phenomenon will be observed in the following simulation of the proposed generator.

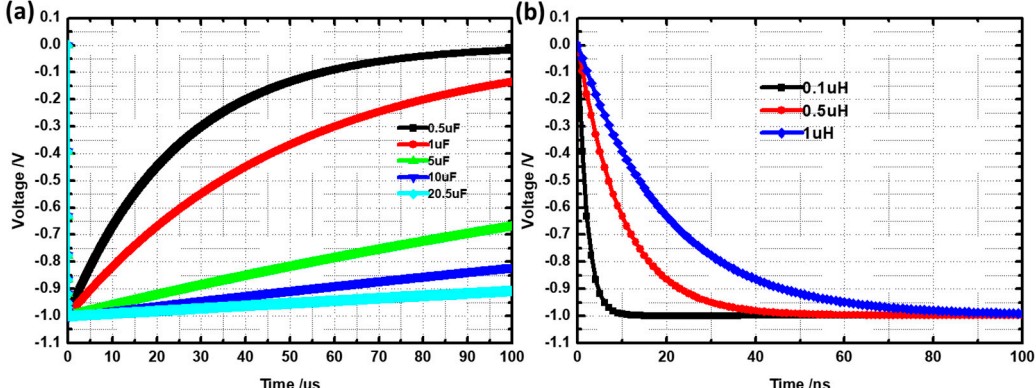

**Figure 2.** (**a**) The voltages on the load with the various discharge capacitor values: 0.5 μF, 1 μF, 5 μF, 10 μF and 20.5 μF under L = 0.1 μH and R = 50 Ω and (**b**) the raising edges of the pulses when the parasitic inductance varies from 0.1 μH to 1 μH under C = 0.5 μF and R = 50 Ω.

Due to the curled-up structure of metal film in a large value capacitor, the additional parasitic inductance would be introduced inevitably and influences exponential function 2 and the rising edge remarkably. As shown in Figure 2b, the rising edge of the pulse spreads wider with the increase of the parasitic inductance even in a small scale. The rising edge has even been extended to 100 ns when the parasitic inductance is increased to 1 μH, which is unacceptable for the nanosecond pulse generation. These results reveal that a large value discharging capacitor is not a good choice for generating nanosecond pulses, but is necessary for producing microsecond pulses.

To achieve the desirable pulse width of 100 ns–100 μs, the contradiction for the choice of discharging capacitance is hard to be solved with a single capacitance. Therefore, the proposed pulse generator has employed a switchable capacitor array.

## 2.2. Implementation of the Pulse Generator

The proposed pulse generator consists of seven modules: PC, high power DC supplier, detecting system, bio-load, low-voltage optic-coupler driver block, high-voltage high-speed switch, and switchable capacitor array, as shown in Figure 3a. The high power DC supplier is TD2200 purchased from Dalian Teslaman Tech. Co., Ltd., Dalian, Liaoning, China and its maximum power is 300 W and the output DC voltage of 0 kV–5 kV, and the detecting system is a digital phosphor oscilloscope DPO5204B, Tektronix Tech. Co., Ltd, Suzhou, Jiangsu, China.

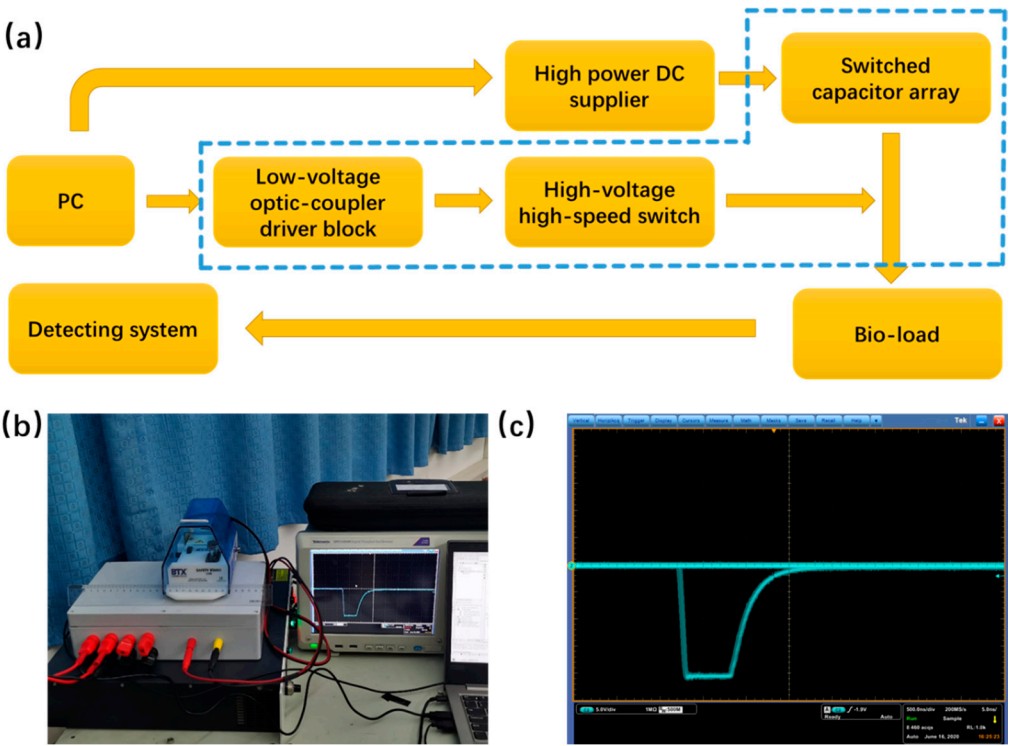

**Figure 3.** (**a**) Schematic diagram of the pulse generator built in seven modules; (**b**) the image of the experimental setup; (**c**) an example of the images of measure results (300 pulses with the pulses width of 500 ns on the swine tissue load).

The three blocks surrounded by the blue dashed line in Figure 3a are the core of this pulse generator and are integrated in a box with a dimension of 288 mm × 210 mm × 80 mm. The low-voltage optic-coupler driver block, shown as Figure 4, is controlled by a PC and drives the high-voltage high-speed switch. The original low-voltage signal is generated by Digilent BASYS2 FPGA board(Digilent Ltd., Pullman, WA, USA) under the control of the program, and its minimum width is 20 ns. Then, original low-voltage signal with the magnitude of 3.3 V is converted to the second signal with the amplitude of 15 V by the high-speed driver, which is MCP14E9 of Microchip Technology Inc., Chandler, AZ, USA, with the bias $V_{c0}$ being set at 15 V and $R_c$ set as 22 Ω. The converted signal actuates the two paralleled optic-coupler microchips (FOD3180s, Fairchild Semiconductor Corporation, Phoenix, AZ, USA), and then drives the high-voltage high-speed switch. The number of microchips depends on the stages of high-voltage high-speed switch, which is set as two in this design. The bias circuit for these two microchips depends on the characteristics of the power MOSFET. The bias voltages $V_{c1}$ and $V_{c2}$ are set at 16 V and the capacitors $C_1$ and $C_2$ are set as 100 pF, respectively, to depress

voltage drop and provide high switch speed. This optic-coupler driver block protects the low voltage module and drives the switch module effectively.

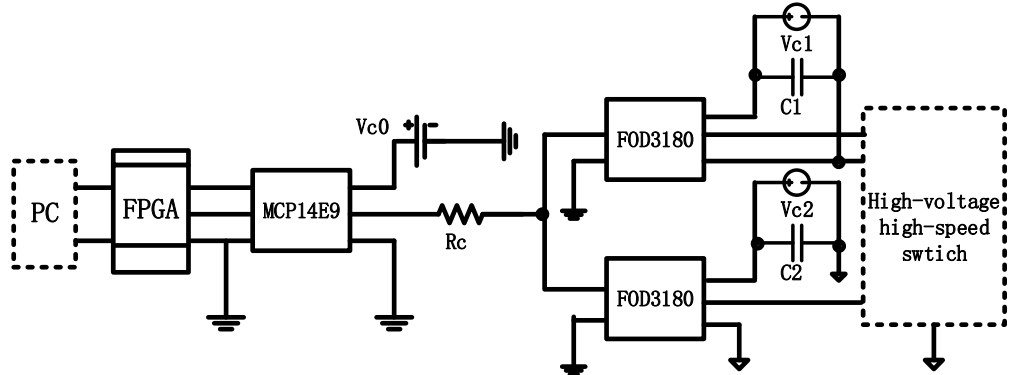

**Figure 4.** Schematic diagram of low-voltage optic-coupler driver block.

As shown in Figure 5a, the high-voltage high-speed SiC MOSFET array is controlled by the low-voltage optic-coupler driver block and connected with the energy storage capacitor array. The unit of this array is built by using SiC MOSFETs SCT2280KE purchased from ROHM Semiconductor (Beijing) Co., Ltd., Beijing, China and transient voltage suppressor diodes 1N5355BG, Semiconductor Components Industries, LLC., Phoenix, AZ, USA. The size of array depends on the voltage of desire pulse and the current influenced by the load. In this design, two stages in series to divide the high voltage of 2 kV while three in parallel to distribute the current flow of 40 A. In addition, the silicon carbide MOSFET has a number of advantages, such as low on-state resistance (static drain source on state resistance is 364 m$\Omega$), high switching speed (the turn-on delay time is 19 ns and turn-off delay time is 47 ns), high breakdown voltage (the drain source voltage is 1200 V) and stable temperature behaviors [35].

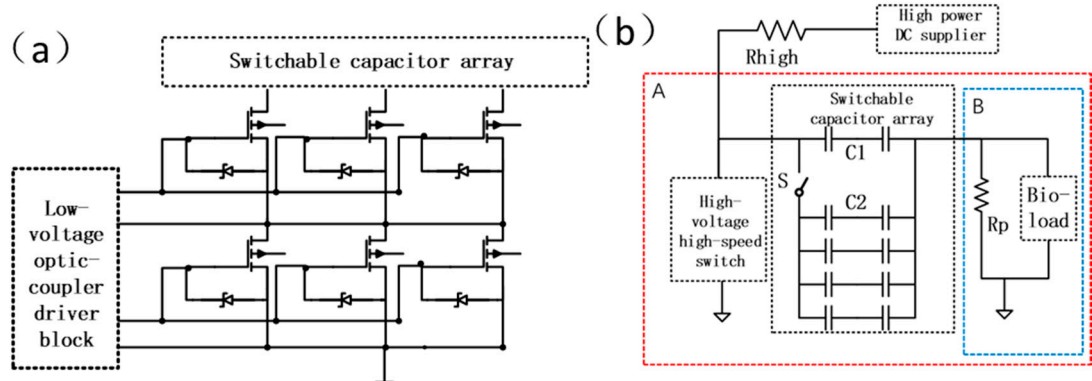

**Figure 5.** Schematic diagram of (**a**) the high-voltage high-speed switch, and (**b**) switchable capacitor array and the operation of the pulse generator.

As shown in Figure 5b, the switchable energy storage capacitor array is fed by a high power DC supplier and the resistor Rhigh set as 10000 $\Omega$ to avoid the DC power supplier overload. In the initial step, the high-voltage high-speed switch is off and bio-load is unconnected, the energy storage capacitor array charges up through Rp and it will play the role of power source in the further steps. In this design, the resistance of Rp is set as 1000 $\Omega$ to balance the need of limiting current in the initial step and the steep raise edge. When the high-voltage high-speed switch is on, the energy storage capacitor array discharges to the load rapidly, leading to the sharp fall edge. The width of the pulses depends on the time of the on-state of the high-voltage high-speed switch. Once the high-voltage high-speed

switch turns off, the load discharge fast through Rp. According to this process, the raise edge depends on electrical properties of bio-load and Rp. At the same time, the energy storage capacitor array is also charged up, while its influence to the pulse is limited by the high resistance of Rhigh.

As in the above description, the intrinsic problem of the capacitor-based discharging system is the difficulty to balance the stored energy for the stability of the peek voltage and sharp rising/falling edges of pulse. To solve this problem, the energy storage capacitor is made of a switchable capacitor array, hereinafter referred to as storage capacitance, needing to be carefully chosen in different situations. The image of the actual experimental setup as shown in Figure 3b. There are two modes in this design: nanosecond pulse mode and microsecond pulse mode. These two modes are switched by the connection of two capacitor arrays C1 and C2. The switch S of this circuit is a manual one. The capacitor array C1 is built by two 940C20W1K-F capacitors (purchased from Cornell-Dubilier electronic, INC., Liberty, SC, USA) with capacitance of 1 μF in series, the total capacitance being 0.5 μF. Further, the capacitor array C2 is composed of 4×2 MKP1848C capacitors (purchased from Vishay China Co., Ltd, Shanghai, China) with capacitance of 10 μF, the total capacitance being 20 μF. In the nanosecond pulse mode, the switch is off, and the first stage capacitor array C1 holds nanosecond pulses (100 ns–2 μs) alone. While in the microsecond pulse mode, the switch is on, and the capacitors C1 and C2 are in parallel and hold microsecond pulses (2μs–100 μs) together.

### 2.3. Simulation and Measurement of the Pulse Generator

The above high voltage width adjusting pulse generator has been modelled using the Multisim 12.0, under different biological load conditions in both nanosecond pulse mode and microsecond pulse mode.

To check the influence of the various biological loads in the nanosecond mode, various resistances of the biological load are set to be: 50 Ω, 100 Ω and 1000 Ω, respectively, while its capacitor value being set as 24 pF invariably. The pulse width was varied from 100 ns to 2 μs, while the high DC voltage was 2 kV. The simulated results are shown in Figure 6.

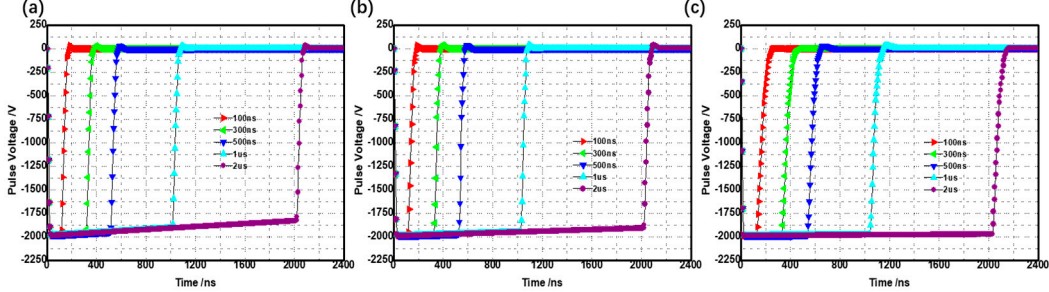

**Figure 6.** The voltages on the different loads with various resistances under the 24 pF capacitance. (**a**) The load resistance is 50 Ω; (**b**) the load resistance is 100 Ω; and (**c**) the load resistance is 1000 Ω.

As shown in Figure 6, the voltage drops at the slightly different rates on the different loads. In addition, the voltage drops more with the increase of the pulse width. When the resistance of the load is 1000 Ω, the pulse has the smallest voltage drop and a rapid rising/falling edge. Because of the different loops of the charge and discharge processes, the rising edge is slower than falling edge. With the decrease of the load resistance, the falling edge slows down while the rising edge steepens. These results reveal that the impedance of the biological load influences the rising/falling edge and the pulse amplitude.

To check the effect of the biological load in the microsecond pulse mode, the load impedances and the bias high DC voltage are set as the same as those in the previous simulation. The discharging capacitor is 20.5 μF (the switch in the capacitor array is on) while the pulse width of the control signal is set as 40 μs, 70 μs and 100 μs. The simulated results are shown in Figure 7.

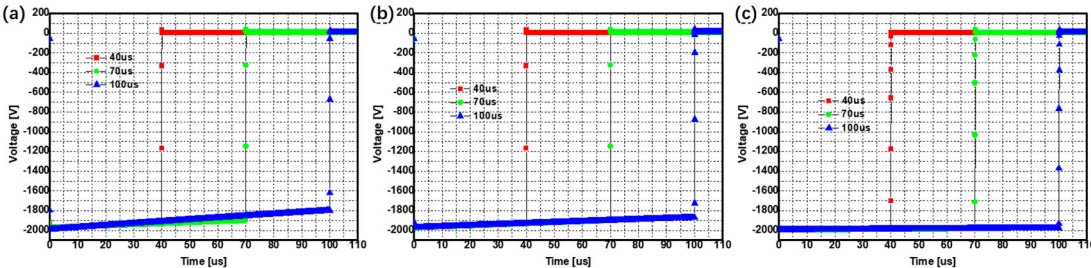

**Figure 7.** The voltages on the three different loads fed by pulses with various pulse widths under the 24 pF capacitance. (**a**) The load resistance is 50 Ω; (**b**) the load resistance is 100 Ω; (**c**) the load resistance is 1000 Ω.

As shown in Figure 7, a large discharging capacitance is essential for holding stable voltages on the all the biological loads in the microsecond pulse mode, especially for the low resistance and the wide pulse width. For the same pulse width, the rising edge slows down and the voltage drop weakens with the increase of the load resistance, which is consistent with the previous analysis and simulation. For the same load, the voltage drop worsens with extending pulse width. These results show the necessity of having a discharging capacitor with the large capacitance in the microsecond pulse mode.

The in-vitro loads are introduced in the experimental testing of the proposed high voltage pulse generator. The first load was a fresh swine tissue hold by a caliper electrode with a gap of 2 mm, the capacitance being around 24 pF and the resistance around 1000 Ω. The caliper electrode is caliper electrode 384 purchased from BTX Co., Ltd. The second load is a cell suspension ($3 \times 10^6$ cells/mL, 5 μL) loaded into the cuvette with a gap of 1 mm, the resistance been around 100 Ω and the capacitance being around 90 pF. The cuvette mounted on a safety stand corrected with the system. The cuvette and safety stand purchased from BTX Co., Ltd. In these two modes, the bias high DC voltage was 2 kV. In the nanosecond pulse mode, the pulse width of the control signal was varied from 100 ns to 2 μs, and the measured results are shown in Figure 8. In the microsecond pulse mode, the pulse widths of the control signal were set at 70 μs and 100 μs, respectively, and the measured results are shown in Figure 9.

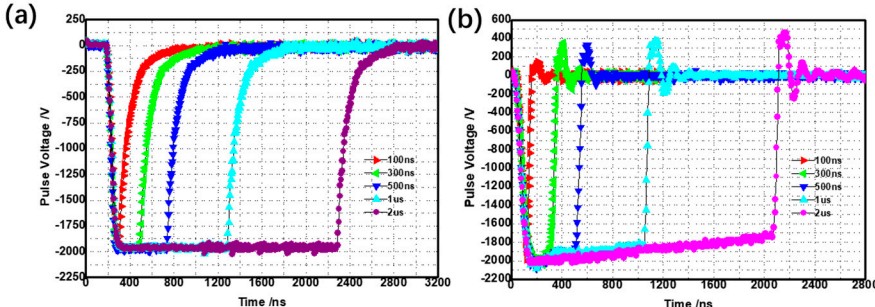

**Figure 8.** The voltages on the in-vitro loads with different widths in the nanosecond pulse mode. (**a**) The swine tissue load; (**b**) the cell culture cuvette load.

In Figure 8, in the nanosecond pulse mode, the pulse widths are 100 ns (red right triangular), 300 ns (green left triangular), 500 ns (blue lower triangular), 1 μs (cyan upper triangular)) and 2 μs (purple dot), respectively. The accuracy of pulse amplitude measurement is 0.1 V and it of pulse length measurement depends on pulse width: 2 ns for the pulses with the pulse width of 100 ns–300 ns and 5 ns for the pulses with the pulse width of 500 ns–2μs. The forms of generated electric field on the two loads are almost uniform in the main area. The mean electric field intensities of the swine tissue load is 10 kV/cm, and it of the cell culture cuvette load is 20 kV/cm. The max duty cycle is 0.0024. For all the conditions, the rising edge is wider than the fall edge, which is caused by the characteristics of the SiC MOSFET and the different loops of the charging and discharging processes. There is rapid rising edge

for all the pulse widths under the low capacitance of 0.5 μF, which is consistent with the analysis in Figure 2. Compared with Figure 8a,b, the voltage drop of the swine tissue load is weaker than it of the cell culture cuvette load for all the pulse widths, which was caused by the different resistances and is consistent with the theoretical analysis. The pulse tail on the cell culture cuvette load, which increases with the pulse width, is caused by the impedance mismatch between the pulse generator and the load. As shown in Figure 3c, this generator possesses relatively high pulse-to-pulse stability. These results verify that this generator is capable of generating intense nanosecond pulse electric fields with the width of 100 ns–2 μs in the nanosecond mode.

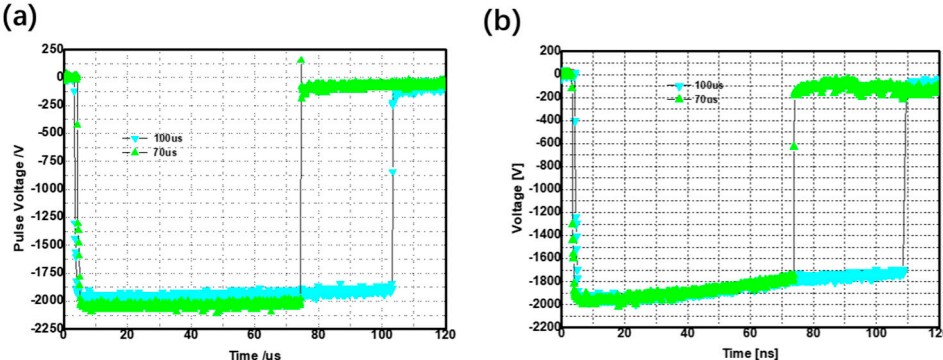

**Figure 9.** The voltages on the in-vitro loads with different widths in the microsecond pulse mode. (**a**) The swine tissue load; (**b**) the cell culture cuvette load.

In Figure 9, the pulse widths are 100 μs (cyan lower triangular) and 70 μs (blue upper triangular), respectively. The accuracy of pulse amplitude measurement is still 0.1 V and it of pulse length measurement is 0.1 μs. The forms and peak electric field intensities of generated electric field are similar to them in the nanosecond pulse mode. The max duty cycle is 0.1. In microsecond pulse model, the falling edge is shorter than the raising edge, which is in contrast of nanosecond pulse mode. It is caused by the additional parasitic inductance, which is introduced by the large capacitor, widening rising edge according to the above analysis. The increasing pulse width leads to aggravating slight voltage drop to an acceptable level, which is consistent with the previous analysis and simulation. Compared with Figure 8a,b, the voltage drop of the swine tissue load is weaker than it of the cell culture cuvette load for all the pulse widths, which is similar to it in the nanosecond pulse mode. These results verify that this generator is capable of generating intense microsecond pulse electric fields with the width of 2 μs–100 μs in the microsecond mode.

By using the developed generator, a preliminary in-vitro experiment has been conducted and reported in [36]. The experiment reveals that nanosecond pluses process a stronger ability to trigger apoptosis compared with microsecond pluses.

## 3. Summary and Future Work

A compact high voltage pulse generator based on a switchable capacitor array and a SiC MOSFET array has been proposed and investigated through analytical calculation, circuit simulation and experimental verification to achieve a large width adjusting range. The developed high-voltage pulse generator is capable of producing the pulses with a voltage range of 0 kV–2 kV, and a pulse width range of 100 ns–100 μs. This generator is capable of meeting many related needs of the current popular biophysical tumor treatment with intensive pulsed electric fields.

With the developed pulse generator, the combined tumor treatment of RE, IRE, nanosecond pulse stimulation and their potential complementary effect will be studied in the future. For the clinical application, the electrical safety test should be implemented, including the over-current and over-voltage safety solutions/measures.

**Author Contributions:** Conceptualization, X.C.; methodology, X.C. and X.R.; validation, X.R.; formal analysis, X.R.; investigation, X.R.; writing—original draft preparation, X.R.; writing—review and editing, X.C., J.Z., B.Z., and Y.A.; supervision, X.C. and J.Z.; project administration, X.C. and J.Z. All authors have read and agreed to the published version of the manuscript.

**Funding:** This research was funded by NSAF under grant No. U1930127, Natural Science Foundation of China under grant No. 61827806, and Fundamental Research Funds for the Central Universities under grant No. ZYGX2019J013.

**Conflicts of Interest:** The authors declare no conflicts of interest.

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
