# Peer review of "Design of a High Voltage Pulse Generator with Large Width Adjusting Range for Tumor Treatment"

_electronics, doi:10.3390/electronics9061053_

Round 1

Reviewer 1 Report

.

Author Response

We acknowledge the reviewer’s comments and have made revision to our manuscript accordingly, the revised texts being highlighted in red. Our responses are summarized as follows.

  1. Formatting – please follow exactly the formatting style when you write the manuscript, please refer to the manuscript template. There are some mistakes like off margin figures (Fig3, fig.5 and Fig.7), sub-titles etc.

Response:

Sorry for the formatting mistakes, Fig3, fig.5 and Fig.7 and sub-titles have been corrected.

  1. The reviewer also found typos in the write up, ie, ‘exponential component1’in section 2.1, ‘capacitors with capacitance of 1μF in serious’ in section 2.2. Please revised these and carefully check other typos also.

Response:

Sorry for the typos, all of them have been revised:

‘exponential component1’has been replaced by ‘exponential function1’;

‘capacitance of 1μF in serious’ has been replaced by ‘capacitance of 1μF in series’;

‘is consist with’ has been replaced by ‘is consistent with’;

‘can’ has been replaced by ‘is capable of …ing‘;

‘the in-vivo load’ has been replaced by ‘the in-vitro load’.

  1. In Introduction, please give justification and provide references for the pulse width of 100 us. References from medical research works may strengthen the justification.

Response:

We thank the reviewer for this comment. We have mentioned the medical research work in three research groups in Ref [9],[11],[16]. Their work show that the pulses with the pulse width of 100ns-100µs are capable of treating tumors based on three different biological effects. In addition, three current commercial devices have been mentioned in Ref [10],[12],[17] to provide the references for the choice of the pulse width adjusting range in the first paragraph in Introduction.

  1. Equation – the detail derivation of Eq.3 and Eq.4 can be included in appendix, it is good for other people to understand how the eigenvalues are evaluated. Please also provide equations for determining time constants of 25 us and 2 ns.

Response:

Sorry for making the reviewer confuse about the eigenvalue evaluation. The eigenvalue evaluation is based on the Viete theorem. The details have been added in line 112-116 in Section 2.1.

The equations for determining time constants have also been provided in Eq. 8 and line 130-131 in Section 2.1.

  1. Please define V(t) in Eq.7 and show it in Figure 1(a).

Response:

Sorry for making the reviewer confuse about the definition of V(t). It is the voltage on the load, which has been added in line 126 in Section 2.1 and the Figure 1(a) has been redrawn.

  1. ef1(t) and ef2(t) notations in Eq.7 are quite confusing, they are like ‘ exponent be multiplied by frequency ‘, please consider the appropriate notation.

Response:

Sorry for making the reviewer confuse about the notations. The ef1(t) and ef2(t)  has been revised as ‘ExpFunc1’and ‘ExpFunc2’in Eq.7

  1. In Fig.5(b), how to control switch’S’? Please explain its mechanism for the proposed system.

Response:

The initial purpose of this design is generating pulses with same parameters in one time. Based on the initial purpose and the consideration of cost and size, the control switch is only one manual switch. It is upgrading, and will be replaced by an IGBT. The choice of switch has been added in line 217 in Section 2.2.

  1. The authors can further improve the manuscript by discussing their future direction.

Response:

We thank the reviewer for this comment. The future work includes the study of RE, IRE, nanosecond pulse stimulation and their potential relationships and the upgrading of the generator, which has been given in the second paragraph in Summary and future work.

Reviewer 2 Report

The manuscript titled “Design of a high voltage pulse generator with large width adjusting range for tumor treatment” by Rao et al. is an interesting work. After reviewing the manuscript, I have the following comments for the authors:

  1. In the abstract section, the authors should provide a little more information on the specifications of the designed pulse generator apart from the adjustable pulse width and amplitude range in terms of PWM frequency, duty cycle, number of output points, number of output channels, etc.
  2. In the lines 14 and 15 in the abstract, the authors state that in order to address the biological effects, the required pulse widths vary from nanoseconds to hundreds of microseconds. However, later in the abstract on line 17, the pulse generator is found to have the maximum pulse width of 100 microseconds. One wonders from these statements as to how many of those biological effects can be addressed seeing that typically pulse widths of hundreds of microseconds (clearly more than 100 microseconds) are required.
  3. In the abstract, the authors should also mention about any prior work on the design and implementation of a similar pulse width generator and how their design is better.
  4. On lines 37 and 38, “… with the advantages of drug-free and non-thermal, but its treatment area is limited (2-3cm)” should be “… with the advantages of drug – free and non – thermal treatment, but its area is limited (2-3cm)”.
  5. On lines 40, 41 and 42, it seems that the authors want to convey the idea of combining the two techniques of using short pulses and microseconds pulses to target tumor treatment. They should consider rephrasing the statement.
  6. In the Introduction section in the first paragraph between lines 26 and 44, where the authors describe the different biological effects, it is suggested that the authors also include the pulse widths required for this biological effects to connect that information to the varying pulse widths provided by the different generators in the subsequent paragraphs. That way the readers will have much better understanding of the different designs of the generators towards targeting different biological effects.
  7. In section 2.1 on the analysis of the capacitor – based discharging circuit, lines 83 to 113, it is unclear which exact exponential forms, referred to as 1 and 2 in the figure 1b, are the authors referring to. It is suggested that the authors insert a statement clarifying this.
  8. On line 140, it should be figure 3a instead of 2a.
  9. Figure 3b shows the photo of a 3 core unit box and figures 4 and 5 shows the schematic diagrams of the three main components of this unit box. Adding an image of the actual experimental setup would definitely add value to this section.
  10. In section 2.3 on line 206, 207, the authors mentioned that the influence of various biological loads were tested using the resistance values of 50, 100 and 1000 ohms, capacitor value of 24 pF and the pulse width of 100ns to 2 microseconds with a high DC voltage of 2 kV. It is suggested that the authors include some of the specific biological effects that they are targeting in testing this pulse generator. For e.g. they can mention that the irreversible electroporation, where the modeled resistance value of so much, capacitor value of so much and the required pulse width of so much was tested using the 2 kV DC voltage supply. They could even take a number of biological effects indicating a range of these values for testing purposes.
  11. On line 235, the authors state that they tested out their designed pulse generator on fresh swine tissue held between the caliper electrodes. It is suggested that the authors include the image of the experimental setup and explain what do they mean when they say that it was an in – vivo load. Typically in – vivo means that the system would reside inside the organism’s physical body. If the said experiment was done on a tissue using a pair of calipers, that would be termed as an in – vitro study.
  12. On lines 246 ad 257, the word is ‘consistent’ not ‘consist’.
  13. Conclusion needs to be little more robust in terms of the performed testing of the pulse generator. It is lacking the data from the experiment on the swine tissue especially in light of the last statement on lines 266, 267 that it is capable of meeting “most” needs of the current popular biophysical tumor treatment with intensive pulsed electric fields.
  14. There are minor grammatical and spell checks that the authors should do.

Author Response

We thank the reviewer for the comments and have made revision to our manuscript accordingly, the revised texts being highlighted in red. Our responses are summarized as follows.

  1. In the abstract section, the authors should provide a little more information on the specifications of the designed pulse generator apart from the adjustable pulse width and amplitude range in terms of PWM frequency, duty cycle, number of output points, number of output channels, etc.

Response:

Thank the reviewer for this comment. The characters including repetition rate and voltage drop have been added in line 17-18 in abstract.

  1. In the lines 14 and 15 in the abstract, the authors state that in order to address the biological effects, the required pulse widths vary from nanoseconds to hundreds of microseconds. However, later in the abstract on line 17, the pulse generator is found to have the maximum pulse width of 100 microseconds. One wonders from these statements as to how many of those biological effects can be addressed seeing that typically pulse widths of hundreds of microseconds (clearly more than 100 microseconds) are required.

Response:

Sorry for making the reviewer confused about the choice of the pulse width adjusting range. IRE and cell apoptosis is stimulated by pulses with the pulse width below 100μs. RE can be stimulated by pulses with the pulse width above about 10μs. But in our further study, the thermal effect should be avoided, so the upper limit of the pulse width was set as 100μs.

To verify that the pulses with the pulse width of 100ns-100µs is capable of treating tumors based on three different biological effects, three clinical works have been mentioned in Ref [9],[11],[16]. In addition, the three current commercial devices have been mentioned in Ref [10],[12],[17] for show the needs of the pulse width for these three biological effects in the first paragraph in Introduction.

  1. In the abstract, the authors should also mention about any prior work on the design and implementation of a similar pulse width generator and how their design is better.

Response:

We thank the reviewer for this comment. The introduction of switchable capacitor array and its unique advantage of a large adjustable pulse width have been highlighted in line 19-21 in abstract.

  1. On lines 37 and 38, ‘… with the advantages of drug-free and non-thermal, but its treatment area is limited (2-3cm)’should be ‘… with the advantages of drug – free and non – thermal treatment, but its area is limited (2-3cm)’.

Response:

We are sorry for the typo, it has been corrected.

  1. On lines 40, 41 and 42, it seems that the authors want to convey the idea of combining the two techniques of using short pulses and microseconds pulses to target tumor treatment. They should consider rephrasing the statement.

Response:

These tumor therapies being triggered by the short width pulsed electric field with different pulse width possess unique advantages and disadvantages. For example, IRE is a nice ablation method without immune effect, while nanosecond pulse stimulation is immune therapy with limited curative effect. Therefore, they may be combined as a complementary therapy.

Of course, it is just a hypothesis for now and the goal of this paper is the introduction of the generator design. The statement has been rephrased in line 54-57 in Introduction.

  1. In the Introduction section in the first paragraph between lines 26 and 44, where the authors describe the different biological effects, it is suggested that the authors also include the pulse widths required for this biological effects to connect that information to the varying pulse widths provided by the different generators in the subsequent paragraphs. That way the readers will have much better understanding of the different designs of the generators towards targeting different biological effects.

Response:

We thank the reviewer for this comment. We have indicated the corresponding biological effects for every mentioned generators in Introduction. In addition, three current commercial device for different biological effects have been added in Ref [10],[12],[17] for making the readers much better understanding of the different designs of the generators towards targeting different biological effects in Introduction.

  1. In section 2.1 on the analysis of the capacitor – based discharging circuit, lines 83 to 113, it is unclear which exact exponential forms, referred to as 1 and 2 in the figure 1b, are the authors referring to. It is suggested that the authors insert a statement clarifying this.

Response:

We are sorry that make the reviewer confused, the statement of two exponential functions has been inserted in the caption of Figure1 and in line 134-135 in Section 2.1. And to make it better understand, ef1(t) and ef2(t)  has been revised as ‘ExpFunc1’and ‘ExpFunc2’in Eq.7

  1. On line 140, it should be figure 3a instead of 2a.

Response:

We are sorry for the typo, it has been corrected.

  1. Figure 3b shows the photo of a 3 core unit box and figures 4 and 5 shows the schematic diagrams of the three main components of this unit box. Adding an image of the actual experimental setup would definitely add value to this section.

Response:

We thank the reviewer for this comment, and the Figure 3b has been replaced as suggested.

  1. In section 2.3 on line 206, 207, the authors mentioned that the influence of various biological loads were tested using the resistance values of 50, 100 and 1000 ohms, capacitor value of 24 pF and the pulse width of 100ns to 2 microseconds with a high DC voltage of 2 kV. It is suggested that the authors include some of the specific biological effects that they are targeting in testing this pulse generator. For e.g. they can mention that the irreversible electroporation, where the modeled resistance value of so much, capacitor value of so much and the required pulse width of so much was tested using the 2 kV DC voltage supply. They could even take a number of biological effects indicating a range of these values for testing purposes.

Response:

As the reviewer suggested, the various impedance will influence the voltage on the load, which may influence the biological effects. The resistances of bio-loads vary from 50 ohms to 1000 ohms, and to verify this influence. The measurement results on the cell culture cuvette load has been added in the fifth, sixth and seventh paragraphs in Section 2.3, which is another typical bio-load.

We have studied IRE and nanosecond pulse stimulation on B16 cells with this developed generator and reported our work in a conference paper [36], which was added at the end of Section 2.3.

To further indicate the width range for the different biological effects, three clinical cases and three current commercial device has been added in Ref [9]-[12], [16], [17] in first paragraph in Introduction.

  1. On line 235, the authors state that they tested out their designed pulse generator on fresh swine tissue held between the caliper electrodes. It is suggested that the authors include the image of the experimental setup and explain what do they mean when they say that it was an in – vivo load. Typically in – vivo means that the system would reside inside the organism’s physical body. If the said experiment was done on a tissue using a pair of calipers, that would be termed as an in – vitro study.

Response:

The reviewer is right in suggesting that the swine tissue held between the caliper electrodes is a simulation of in-vivo load, instead of a truly in-vivo load. The related statements have been corrected.

  1. On lines 246 ad 257, the word is ‘consistent’ not ‘consist’.

Response:

We are sorry for the typos, they have been corrected.

  1. Conclusion needs to be little more robust in terms of the performed testing of the pulse generator. It is lacking the data from the experiment on the swine tissue especially in light of the last statement on lines 266, 267 that it is capable of meeting ‘most’needs of the current popular biophysical tumor treatment with intensive pulsed electric fields.

Response:

As in the reviewer’s comments, we have referred to an in-vitro experimental work being reported in BMES 2019 [36] to verify the performance of this generator in the end of Section 2.3. While the in-vivo experiment is still in progress, the measurement on the swine tissue shows a nice performance of the generator, producing the clean pulses on bio-loads. The three clinical cases mentioned in Ref [9], [11], [16] have further justified the need of these pulses.

The reviewer is right in suggesting that the range of ‘most’ is hard to defined, and we have tuned down ‘most’ as ‘many relatedin first paragraph in Summary and future work

  1. There are minor grammatical and spell checks that the authors should do.

Response:

We are sorry for the typos, all of them has been revised:

‘exponential component1’ has been replaced by ‘exponential function1’;

‘capacitance of 1μF in serious’ has been replaced by ‘capacitance of 1μF in series’;

‘can’ has been replaced by ‘is capable of  …ing‘.

Reviewer 3 Report

The manuscript reports original results from design and development of a lab-made electric pulse generator. Its main feature is the usage of a MOSFET switching array and a capacitor-based discharging system. The principal design and its realization had been intended to get extended range of pulse parameters – output voltage and pulse width. The proposed system is going to meet wide applicability in various areas of Physical Biomedicine.

The system architecture is clearly presented. It is realized by commercially available devices and electronic components. The design principles and operation mechanisms are credibly analyzed, simulated, realized and experimentally tested. However, presenting results of only one “in-vivo” experiment, the verification of proposed system is not sufficiently performed. Moreover, it is done for a case which is far from the very system application – the “tumor treatment”.

The article fullness and the reproducibility of results will benefit if more characteristics are added, such as: the accuracy of pulse amplitude and pulse length, the pulse-to-pulse stability, the max duty cycle, the form and values of generated electric field, some over-current and over-voltage safety solutions/measures, etc.

It should be noted, that a comparison with other known similar systems, incl. commercially available (AVTECH Electrosystems), is not presented in full. Also, I would recommend to reflect the classical article of W. Jiang et al (DOI: 10.1109/JPROC.2004.829003) and the most recent article of V. Stankevic (DOI: 10.3390/sym12030412).

As a whole, the article could be a good and useful contribution to the area of self-done electronic systems for biomedical applications.

Author Response

We acknowledge the reviewer’s comments and have made revision to our manuscript accordingly, the revised texts being highlighted in red. Our responses are summarized as follows.

  1. However, presenting results of only one ‘in-vivo’ experiment, the verification of proposed system is not sufficiently performed. Moreover, it is done for a case which is far from the very system application – the ‘tumor treatment’.

Response:

We thank the reviewer for this comment, and the measurement results on the cell culture cuvette load has been added to better verify the performance of the proposed system in the fifth, sixth and seventh paragraphs in Section 2.3.

According to the reviewer’s comments, to verify these pulse can treat tumor, three clinical cases mentioned in Ref [9], [11], [16] have been added in Introduction and some conclusions based on our published work [36] have been added in the eighth paragraph in Section 2.3. According to above reasons, we can infer that the performance of the developed generator in the tumor treatment.

2.The article fullness and the reproducibility of results will benefit if more characteristics are added, such as: the accuracy of pulse amplitude and pulse length, the pulse-to-pulse stability, the max duty cycle, the form and values of generated electric field, some over-current and over-voltage safety solutions/measures, etc.

Response:

We thank the reviewer for this comment. The characteristics has been added as suggested in the sixth and seventh paragraphs in Section 2.3 and Figure3 (c), which include the accuracy of pulse amplitude and pulse length, the pulse-to-pulse stability, the max duty cycle, the form and values of generated electric field.

Sorry for that the electrical safety test is still in progress, and we will finish over-current and over-voltage safety solutions/measures in the future following the reviewer’s suggest, and these future work has been discussed in second paragraph in Summary and future work.

  1. It should be noted, that a comparison with other known similar systems, incl. commercially available (AVTECH Electrosystems), is not presented in full. Also, I would recommend to reflect the classical article of W. Jiang et al (DOI: 10.1109/JPROC.2004.829003) and the most recent article of V. Stankevic (DOI: 10.3390/sym12030412).

Response:

We thank the reviewer for this comment, and it is very important that comparison of the commercial devices. We compared with four commercial devices: BTX AgilePulseTM System of BTX Co., Ltd, Angio-Dynamics NanoKnife System of Angio-Dynamics Co., Ltd, CellFX™ System of Pulse Biosciences Co., Ltd and AVRH-2-B pulse generator of Avtech Electrosystems Ltd in first and third paragraphs in Introduction.

We thank the reviewer for the two articles. The article of W. Jiang et al shows the advantages and disadvantages of the different configurations and related solid switch, which has been mentioned in Ref [32]. The article of V. Stankevic is a nice generator for electroporation, and it also reveals that the important role of switchable capacitor array in the design with a large adjustable pulse width, which has been mentioned in Ref [34].

Round 2

Reviewer 1 Report

Thank you